biomechanics/fluid mechanics

two-phase flows, flight, hummingbirds, insect, swifts

**Author for correspondence:**
Victor M. Ortega-Jimenez
e-mail: ornithopterus@gmail.com

# Natural barriers: waterfall transit by small flying animals

Victor M. Ortega-Jimenez[1], Eva C. Herbst[2], Michelle S. Leung[3] and Robert Dudley[1,4]

[1]Department of Integrative Biology, University of California, Berkeley, CA 94720 USA
[2]Palaeontological Institute and Museum, University of Zurich, Switzerland
[3]University of California, San Francisco School of Medicine, San Francisco, CA 94143 USA
[4]Smithsonian Tropical Research Institute, Balboa, Republic of Panama

VMO-J, 0000-0003-0024-5086; ECH, 0000-0003-3640-9695

Waterfalls are conspicuous geomorphological features with heterogeneous structure, complex dynamics and multiphase flows. Swifts, dippers and starlings are well-known to nest behind waterfalls, and have been reported to fly through them. For smaller fliers, by contrast, waterfalls seem to represent impenetrable barriers, but associated physical constraints and the kinematic responses of volant animals during transit are unknown. Here, we describe the flight behaviour of hummingbirds (the sister group to the swifts) and of various insect taxa as they fly through an artificial sheet waterfall. We additionally launched plastic balls at different speeds at the waterfall so as to assess the inertial dependence of sheet penetration. Hummingbirds were able to penetrate the waterfall with reductions in both their translational speed, and stroke amplitude. The body tilted more vertically and exhibited greater rotations in roll, pitch and yaw, along with increases in tail spread and pitch. The much smaller plastic balls and some flies moving at speeds greater than 2.3 m s$^{-1}$ and 1.6 m s$^{-1}$, respectively, also overcame effects of surface tension and water momentum and passed through the waterfall; objects with lower momentum, by contrast, entered the sheet but then fell along with the moving water. Waterfalls can thus represent impenetrable physical barriers for small and slow animal fliers, and may also serve to exclude both predators and parasites from nests of some avian taxa.

## 1. Introduction

Waterfalls are majestic cascades created by rivers, streams, surface runoff, rainstorms and melting ice. These unsteady multiphase flows, dynamics of which are governed by gravity and surface tension, accrue kinetic energy as they accelerate downward, and

also gradually break up into droplets while falling [1]. Such flows probably represent a substantial environmental barrier for smaller volant taxa, although various bird taxa (dippers, starlings and swifts [2–4]) often construct their nests near or behind waterfalls, and have been reported to fly through them. Those physical factors that prevent smaller fliers from passing through waterfalls, as well as associated biomechanical responses, have not been described.

Small animal fliers, such as hummingbirds and insects, seem behaviourally to avoid waterfalls and tend not to cross them. However, these taxa are commonly exposed to severe precipitation events which are two-phase flows. For example, hummingbirds [5] and mosquitoes [6] can easily compensate for wetting and impact forces produced by strong rain. Nevertheless, the momentum of smaller fliers may be insufficient to travel through waterfalls. Small dipteran insects (approx. 1–100 mg) and many hummingbirds (approx. 5 g), for example, are typically orders of magnitude less massive than swifts (approx. 40 g). Even seabirds such as diving petrels, which are known to fly through large waves during rough ocean conditions, are relatively fast and heavy (mass of approx. 150 g [7]). Therefore, scale-dependencies can influence transit through either waterfalls or dynamically comparable two-phase flows.

Here, we investigated the effects of scale and biomechanics on the ability of small fliers to pass through waterfalls. First, we analysed the effects of flight through an artificial water curtain on the wing and body kinematics of Anna's hummingbirds. Hummingbirds (a lineage closely related to swifts) can fly and nectar-feed in heavy aerial turbulence [8,9] and in heavy rain [5]. They also occasionally nest at branch tips near small waterfalls in the Neotropics (R. Dudley 1998 and 2006, personal observation), and bathe within pools at the base of waterfalls [10], so at least occasionally may fly through complex water flows. For comparative purposes, we also assess the ability of various free-flying insects of different sizes to pass through the artificial waterfall. Finally, we launched small styrofoam balls at different speeds against the moving water curtain, in order to characterize those inertial factors possibly influencing successful transit.

# 2. Material and methods

## 2.1. Study taxa

Four adult male Anna's hummingbirds were captured on the campus of the University of California, Berkeley, CA, USA; all birds were released into the wild at the point of capture following experiments. Three taxa of dipteran insects were also obtained on campus (green bottle flies: *Lucilia* sp. ($N = 7$); house flies, *Musca* sp. ($N = 24$), and one cranefly, *Tipula* sp.); fruit flies (*Drosophila melanogaster*) were obtained from a laboratory colony, but were uniformly unable to fly through the experimental waterfall (see below). Morphological data for the four hummingbirds, including body mass, wing length and the area of one wing, are provided in electronic supplementary material, table S1, along with representative data for single individuals of the aforementioned dipteran taxa. We choose the aforementioned taxa because hummingbirds are the sister group to swifts and can be studied experimentally in the laboratory. Dipterans are easily obtained study taxa for flight research. Both groups are commonly subjected to and fly within heavy rain in their natural environments.

## 2.2. Waterfall design, filming and analysis

A curtain waterfall was created using a water jet flowing from a pipe and then passing over a plastic diffuser sheet positioned at the top of an acrylic flight chamber ($60 \times 30 \times 30$ cm; see figure 1). The water jet was continuously generated by a submersible pump (Shysky Tech DC50A-1235A, DC50B-24130A) positioned within a water reservoir ($30 \times 30 \times 15$ cm) directly beneath the flight chamber. The flow rate per horizontal length of the water curtain was approximately 2 ($l\,min^{-1}$) $\times$ cm, with a thickness of approximately 3 mm in the approximate region where hummingbirds penetrated the waterfall. We used a force transducer (Nano17, ATI Industrial Automation, Apex, NC, USA) to measure the hydrodynamic force per unit length over time as produced on a small plastic plate ($2.5 \times 2.7$ cm) positioned perpendicular to flow; this force averaged a value of $1.3 \times 10^{-2}\,N\,cm^{-1}$ (range: $1.23$–$1.34 \times 10^{-2}\,N\,cm^{-1}$). Two synchronized high-speed cameras (HiSpec, Fastec Imaging) operated at 500 frames $s^{-1}$ were positioned lateral to and in front of the waterfall to record transits (figure 1).

Hummingbirds were trained over multiple days to feed volitionally from a nectar feeder positioned on one side of the flight chamber, with the waterfall turned off. We then placed a bird within the chamber and intermittently turned on the waterfall, such that the hummingbird had to pass through the water curtain to return to the perch after feeding. For comparison, we also filmed the same individual returning to the perch

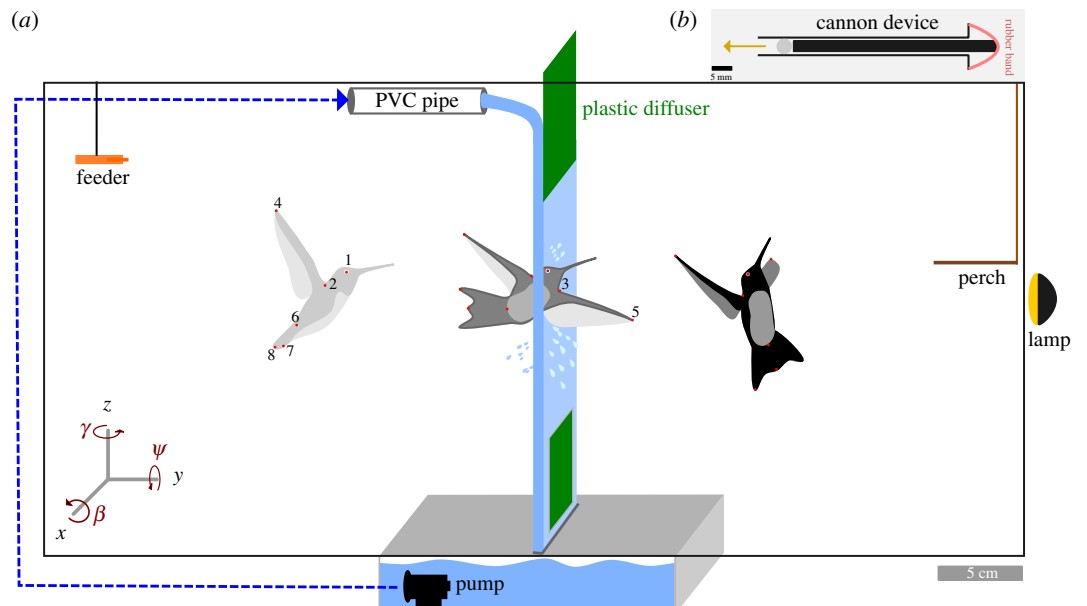

**Figure 1.** (*a*) Experimental configuration of the artificial waterfall, and cartoon of a transiting hummingbird; red dots indicate particular body landmarks mentioned in text: (1) right eye, (2) right-wing base, (3) left-wing base, (4) right-wing tip, (5) left-wing tip, (6) base of tail, (7) tip of outer right tail rectrix, (8) medial tip of the tail. (*b*) Cannon used to launch styrofoam balls into the waterfall.

in the absence of a waterfall; video sequences of one such control flight and one waterfall transit for each of four hummingbirds were analysed in detail. We digitized (using the DLTdv digitizing tool for Matlab, http://biomech.web.unc.edu/dltdv/) the positions of the right eye, the base of the tail, the tip of the tail, the tip of the outer right rectrix and the wing tips and shoulders (i.e. the wing roots, when visible; see [8]). Using those digitized points as calibrated in three-dimensional space via a $8 \times 8 \times 8$ cm cube with 32 landmarks, we calculated body pitch as the angle of the vector formed by the eye and base of the tail in the $(y, z)$ global plane, yaw as the angle of the vector formed by the eye and base of the tail in the $(x, z)$ plane, roll as the angle of the vector formed by both shoulders in the $(x, y)$ plane, tail pitch as the angle between the base and tip of the tail, and tail spread as the angle between the vector of the base-tail tip and the base-tip of the outer right rectrix. Stroke amplitude was calculated as the angle formed by the wingtip and the shoulder at the two wingbeat extremes, and flapping frequency as the inverse of the time required to complete one full wingbeat. Flight speeds and accelerations were calculated from the first and second derivatives of the mean square error (MSE) quintic spline function (see [11]) based on the position of the eye through time (see figure 1). Mean values for each of the aforementioned parameters were calculated for approximately 50 ms prior to wing contact with the waterfall, for approximately 80 ms while birds were in the waterfall and for 150 ms following either wing or body contact with the waterfall. Flight trajectories were nominally perpendicular with respect with the water sheet, and thus we report average three-dimensional speed and acceleration values for the frame of reference as shown in figure 1.

To elicit flight of insects through the waterfall, we placed an incandescent light (250 W) directly outside of the chamber (see figure 1). Insects were released at the opposite side of the flight chamber while the waterfall was running, and in many cases then flew directly toward the light and thus entered the waterfall. For each recorded flight sequence, we digitized the positions of the head and the tip of the abdomen through time, and then calculated values of body pitch. Body speeds were calculated as the average of the first five velocity values calculated using the first derivative of the MSE quintic spline of $(x,y,z)$ positional data (i.e. $U_{ini}$) and the horizontal component of the final five speed values ($U_{end}$) obtained for each filmed sequence. The value of $U_{end}$ for insects that failed to cross the waterfall thus corresponds to the near-zero horizontal speed reached while falling during impact. Similarly, body pitch was calculated as the average of the 10 pitch values immediately preceding waterfall transit, and for the 10 values at the end of the recorded positional data.

We also performed a ballistic experiment using small styrofoam balls (4.5 mm diameter) shot at various speeds using a small cannon positioned horizontally such that its aperture was 3 cm from the waterfall. The cannon consisted of a metal tube (5 mm outer diameter, with a length of 10 cm), an

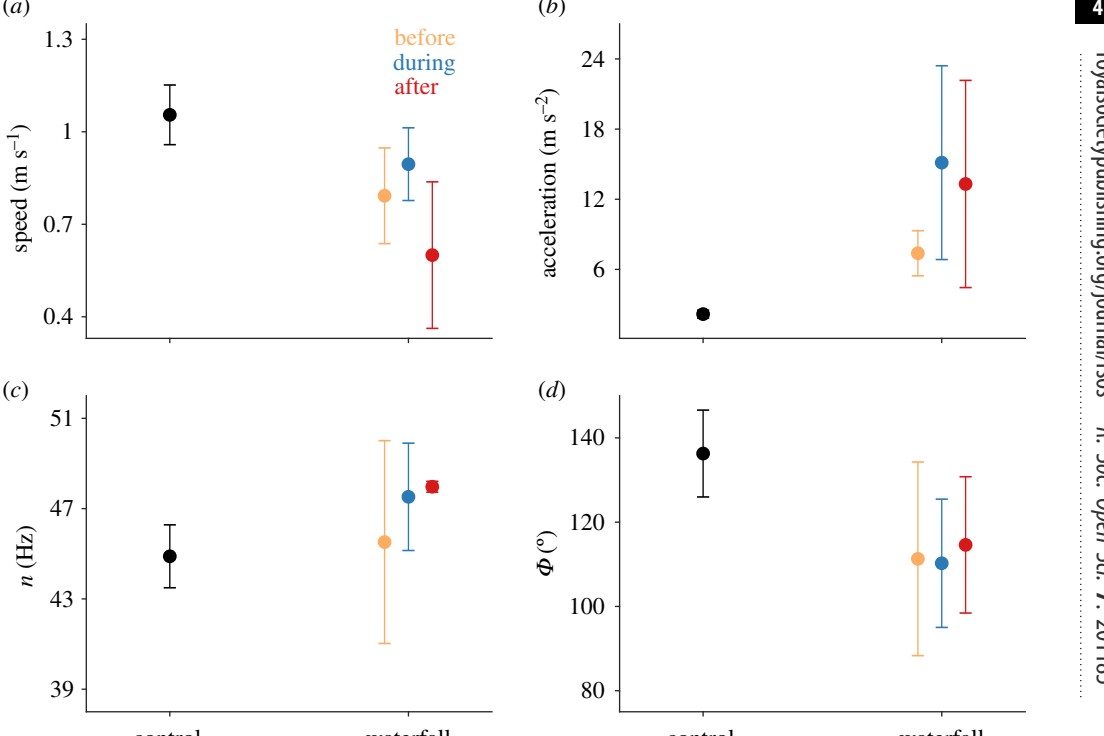

**Figure 2.** (*a*) Speed, (*b*) acceleration, (*c*) stroke frequency and (*d*) stroke amplitude for four hummingbirds flying in still air (black), and before (orange), during (blue), and after (red) crossing the waterfall. Data points represent means ± one standard deviation; see electronic supplementary material, table S2 for statistical results.

inner rod (4.5 mm diameter and 11 cm long) and a rubber band (see figure 1*b*). Using a single high-speed camera positioned laterally, we filmed trajectories of 59 balls (with an average mass of 1.7 mg) shot at speeds from approximately 1 to approximately 9 m s$^{-1}$ into the waterfall. For each such trajectory, we digitized the position of centre of mass for the ball through time. As with insects, ball speeds were calculated as averages of the five first initial values ($U_{ini}$) and final five horizontal values ($U_{end}$) for each recorded sequence.

We used repeated-measures ANOVAs to compare body pitch, roll, and yaw, stroke amplitude, speed, and tail pitch and spread among the control condition of flight in still air and the three time periods when flying through the waterfall (i.e. before, during and after transit). Because of violations of normality and sphericity for data on wingbeat frequency and acceleration, we used a non-parametric Friedman test to compare these measurements among the aforementioned four groups. Pairwise comparisons (either Tukey or Friedman *post hoc* tests) were then used if necessary. Body orientations of insects were compared before and after passing the waterfall using a Wilcoxon paired test; linear regressions were fit between initial speed and final horizontal speeds for all insects, and also for the styrofoam balls launched at the waterfall. All statistical analyses were performed using R v. 3.4.4 [12]. Data are presented as the mean value ± one standard deviation.

## 3. Results

All four hummingbirds passed through the waterfall in less than 100 ms (average duration of 79 ± 10 ms), and showed marked changes in kinematics during and after transit. Three individuals flew through the waterfall sideways, using one wing tip to initially break the water curtain (electronic supplementary material, video S1), but one individual passed through symmetrically with minimal body yaw relative to the waterfall (electronic supplementary material, video S2). Kinematic parameters during control flights and those before, during and after transiting the waterfall differed significantly (figures 2 and 3): body pitch ($F_{3,9} = 16.6$, $p < 0.001$), body roll ($F_{3,9} = 5.5$, $p = 0.02$), body yaw ($F_{3,9} = 8.3$, $p < 0.006$), stroke amplitude ($F_{3,9} = 4.1$, $p = 0.04$), tail pitch ($F_{3,9} = 13.7$, $p = 0.001$), tail spread ($F_{3,9} = 14.7$, $p < 0.001$), speed ($F_{3,9} = 6.4$, $p = 0.01$) and acceleration ($\chi2 = 9.3$, d.f. = 3, $p = 0.02$). Translational speeds were lower post-

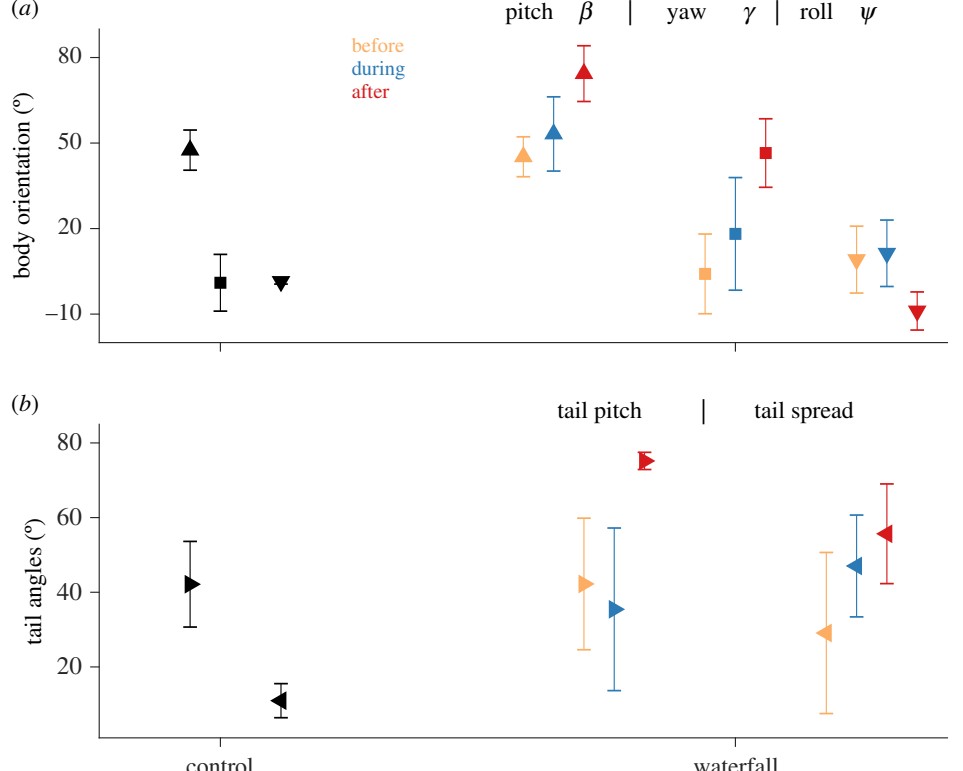

**Figure 3.** (a) Body and (b) tail angles for the same four hummingbirds flying in still air (black), and before (orange), during (blue), and after (red) crossing the waterfall. Symbols in (a) denote pitch (upward triangle), roll (square) and yaw (downward triangle); those in (b) indicate tail pitch (triangle pointing right) and tail spread angle (triangle pointing left). Data points represent means ± one standard deviation.

transit relative to speeds of control flights, whereas accelerations were higher only before crossing the waterfall relative to controls (see figure 2, electronic supplementary material, table S2). Body pitch, roll and yaw, and also tail pitch were significantly higher after birds passed through the waterfall relative to the other three conditions (figure 3, electronic supplementary material, table S2). Tail spread angle was higher after passing through the waterfall in comparison with values both before transit and in control flights; tail spread during crossing was higher than in control flights (see figure 3, electronic supplementary material, table S2). Stroke amplitude was higher during control flights relative to flight after waterfall transit. We did, however, find no significant differences in wingbeat frequency among the four conditions ($\chi^2 = 4.5$, d.f. = 3, $p = 0.25$). Water droplets were also observed to remain attached to the beak and plumage after passing through the waterfall (see electronic supplementary material, video S3).

No fruit flies could fly through the waterfall, but all bottle flies and all house flies moving at speeds greater than 1.6 m s$^{-1}$ could cross the air–water interface (see figure 4). Some large bottle flies with speeds closer to 1 m s$^{-1}$ also were successful in crossing. The crane fly flew at a speed less than 1 m s$^{-1}$ and failed to cross the waterfall; its legs interacted first with the descending water and caused a substantial nose-down pitch down before the body entered the flow. All insects which were able to transit the waterfall demonstrated significant reductions in body pitch (by 59% on average; Wilcoxon signed ranked test, $V = 78$, $p < 0.001$; see figure 5). Some insects retained water droplets on their bodies after passing through the waterfall (electronic supplementary material, video S1). Speed and body pitch for flies that successfully flew through the waterfall were positively correlated (figure 5b). Finally, plastic balls shot directly into the waterfall passed through only when their speed was greater than 2.3 m s$^{-1}$; initial and final speeds for such transits were positively correlated (see figure 6).

## 4. Discussion

This study presents novel biomechanical data for animal fliers challenged by a complex (and to date unstudied) multiphase flow regime. Passing through a waterfall poses substantial challenges to flight

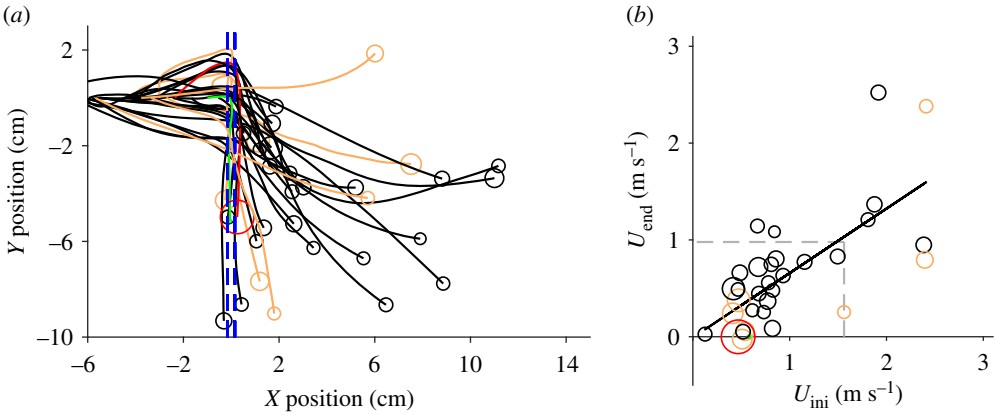

**Figure 4.** (a) Two-dimensional insect trajectories for dipteran insects: bottle flies: *Lucilia* sp. ($N = 7$, orange symbols); house flies, *Musca* sp. ($N = 24$; black symbols), one cranefly, *Tipula* sp. (red symbol), and a fruit fly (*Drosophila melanogaster*; green symbol). Circle size represents relative body size; the waterfall position is represented by dashed blue lines. (b) Body speed ($U_{ini}$) prior to waterfall contact versus horizontal speed ($U_{end}$) following transit through the water fall. Circle sizes represent relative body size and average speed. Grey broken lines indicate the initial speed (approx. 1.6 m s$^{-1}$) below which no insect could pass successfully through the waterfall.

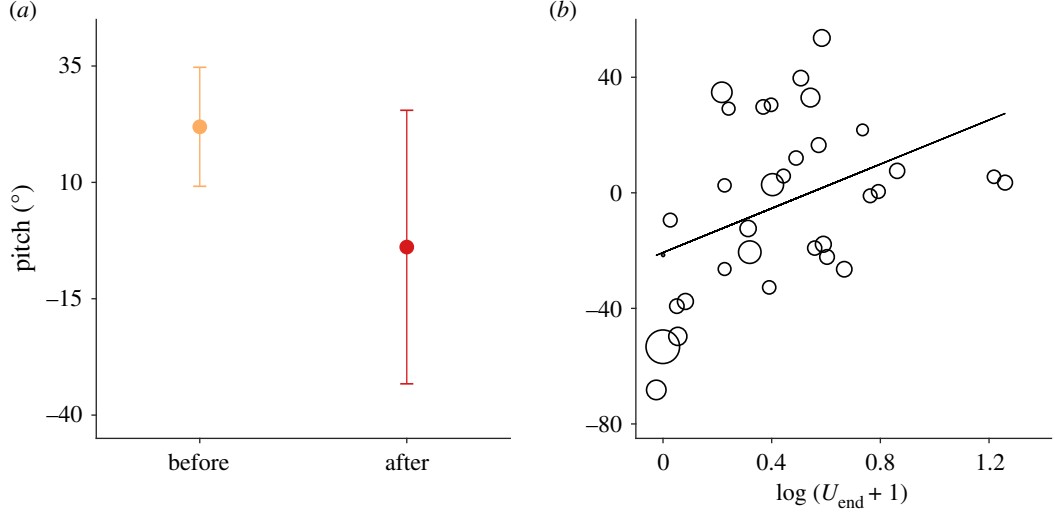

**Figure 5.** (a) Body pitch of flies ($N = 33$) prior to (orange symbols) and after (red symbols) interacting with the waterfall. Data points represent means ± one standard deviation. (b) Body pitch versus log (horizontal speed + 1) following passage through the waterfall; circle sizes represents relative body size. The regression is given by: pitch = 38 log ($U_{end}$ + 1) − 21 ($r^2 = 0.2$, $F_{31} = 6.7$, $p = 0.01$).

control, particularly for small animals. Hummingbirds were successful in this task, but slow-moving insects entered and remained trapped in the downward water flow. Insects that did cross the waterfall typically experienced substantial nose-down pitch and a subsequent downward trajectory, but some individuals recovered their trajectory and even flew upward (see figure 4a). All four hummingbirds, by contrast, rapidly transited the waterfall (albeit with transient disruption to body, wing and tail kinematics), and then flew directly to their perch. Scale-dependence of successful waterfall penetration is thus indicated, and probably derives from effects of multiple physical factors, including surface tension of the water surface, changes in body inertia relative to dynamic loading and torque imposed by the moving water, and possible mass loading post-transit by adhered water. Only hummingbirds and some insects maintained an upward trajectory after crossing the waterfall (electronic supplementary material, figure S1a). In particular, hummingbirds were able to fly at a nominal perpendicular angle (±10°) relative to the water sheet. Entering a waterfall at a negative angle will probably increase downward speed, whereas flying through with an upward speed component can reduce the horizontal speed component and (for an insect) reduce the likelihood of crossing (electronic supplementary material, figure S1b).

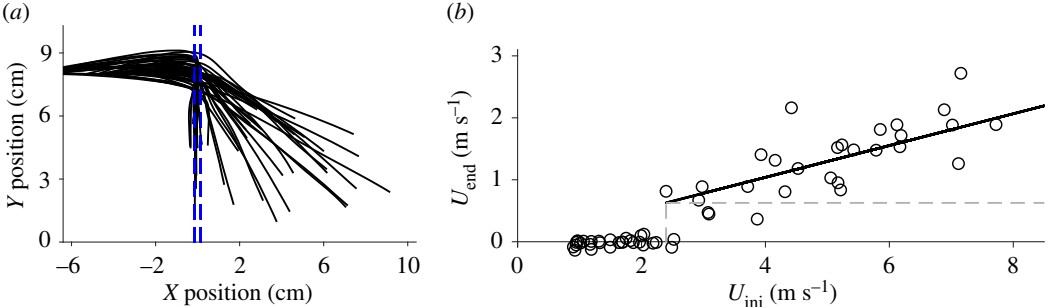

**Figure 6.** (a) Trajectories of styrofoam balls ($N = 59$) launched into the waterfall, the position of which is represented by broken lines in blue. (b) Initial ball speed ($U_{ini}$) versus final speed ($U_{end}$) at the end of the trial. Grey broken lines indicate the initial speed at which balls successfully crossed the waterfall (i.e. $U_{ini} > 2.3$ m s$^{-1}$). The regression is given by $U_{end} = 2.2\ U_{ini} + 2.3$ ($r^2 = 0.56$, $F_{1,28} = 35.7$, $p < 0.001$).

Nevertheless, a high transit speed dramatically increases droplet splashing, which may worsen wetting effects and impact forces on wings (see electronic supplementary material, video S2). Hummingbirds and insects tended to negotiate the water sheet used here at lower speeds relative to launched small balls (see electronic supplementary material, figure S1b).

By launching various sized balls (from approx. 1 to approx. 8 cm diameter and from approx. 1 to approx. 6 g) at similar speeds against a water sheet, we demonstrated that only the larger ones were able to pass through (electronic supplementary material, video S2). This result suggests that swifts, starlings and dippers, all of which are much heavier than hummingbirds and insects, may be able to pass through moving water structures simply because of their greater momentum. Natural waterfalls, although apparently not yet characterized quantitatively on the spatial scales relevant to volant taxa, present much more complex flows than those used here, including substantial spray zones, voids and strongly turbulent water structures. The techniques used by swifts flying through such flows, although highly challenging to study, would now be of interest for future study given the flight behaviours of hummingbirds as described in this study.

Cohesive forces among water molecules maintain sheet flow within a waterfall, and breaking a static water interface requires a quantity of energy per unit area exceeding that of surface tension $\sigma$ (approx. $72 \times 10^{-3}$ J m$^{-2}$). When water striders jump from water, a value of approximately $2\sigma$ is necessary to break the air–water interface [13]. Here, plastic balls with a mass about twice of a fruit fly could transit the experimental waterfall only if their speed exceeded 2.3 m s$^{-1}$, corresponding to a kinetic energy of about approximately $5 \times 10^{-6}$ J. Given a cross-sectional area for the balls of approximately $3.2 \times 10^{-5}$ m$^2$, the area-specific energy of the balls is approximately $160 \times 10^{-3}$ J m$^{-2}$ or about twice the area-specific energy necessary to break the interface. Similarly, flies flying at speeds greater than 1.6 m s$^{-1}$ (mean body diameter of approx. 0.5 cm, mass of 20 mg), travel with sufficient energy per cross-sectional area ($162 \times 10^{-3}$ J m$^{-2}$), to easily break the surface tension. Fruit flies moving here at 0.6 m s$^{-1}$ (or even at maximum speeds of 1.1 m s$^{-1}$ [14]), generated kinetic energy per unit area equal to 0.33–1$\sigma$, and thus could not break through the moving water sheet. Insects such as craneflies experience even greater difficulties, as penetration is initiated by deformable legs or antennae which remain attached to the moving water and result in substantial torque and downward entrainment of the body. The latter outcome agrees with experiments performed on mosquitoes exposed to falling droplets; impacts on their wings and legs can cause marked changes in body orientation, with a recovery time of approximately 100 ms [6]. For much larger animals such as hummingbirds, flapping with high wing tip velocities (approx. 9 m s$^{-1}$) generates high tip forces sufficient to break the water sheet and to pass through.

In addition to breaking surface tension, passage through a waterfall requires momentum to offset the downward inertial loading of the moving water; this effect is also size-dependent. If we assume a hummingbird with a wing span of 12 cm to be positioned with wing tips (and body) immersed in the moving sheet, and assume the static hydrodynamic force per distance to be $1.3 \times 10^{-2}$ N cm$^{-1}$ (as measured on a static plate; see Material and methods), then the downward force will be approximately three times the body weight (approximately the maximum load-lifting capacity of hummingbirds [15]). Hummingbirds confronted with such sudden loading exhibited higher accelerations but slower speeds after passing the waterfall, in comparison with their control flights (figure 2). Furthermore, such a high transient force can be mitigated by flying sideways through the waterfall, as seen in three of the four birds studied here. By contrast, a bottle fly with a span of approximately 2 cm will experience a

downward force two orders of magnitude greater than its weight. By contrast to hummingbirds, insects also generally entered the waterfall head-first, resulting in a nose-down pitching moment that impeded recovery. Faster flight, however, reduces the total hydrodynamic impulse and results in smaller changes in body pitch post-transit (figure 5b).

Although typically hydrophobic, both insect cuticle and avian feathers can become wet during passage through a waterfall. When a two-phase flow collides with a textured surface, splashing ensues which can produce a wetting transition depending on the pressure exerted by the moving water [16]. Here, wing tips of hummingbirds collided with the water at speeds of approximately $9 \, \text{m s}^{-1}$, yielding an impact pressure near 60 kPa (see also [5]) and droplet accumulation on the feathers. Bottle flies can similarly produce splashing because wingtip speeds reach up to approximately $5 \, \text{m s}^{-1}$ [17], but fruit flies probably do not because their smaller wings travel at a much lower value of approximately $1.5 \, \text{m s}^{-1}$ [18]. We observed splashing generated by larger flies, and also noted that some water droplets remained anchored on their bodies. However, wing impact with the waterfall can impede or even prevent flapping, as observed in some flies (electronic supplementary material, video S1). Water attached to the body can also dramatically elevate the effective mass of the system; a 5 mm droplet has a mass of approximately 0.06 g, which is comparable to the mass of a bottle fly. Attached water may also produce a sudden shift in the centre of mass, challenging flight stability and control. For example, a thin plastic plate ($0.05 \times 0.8 \times 3$ cm, mass of 0.08 g) launched through the waterfall results in attachment of droplets to the trailing edge, causing upward pitching after transit (see electronic supplementary material, video S1).

Hummingbirds exposed to rain and to turbulent airflow [9] respond with changes in wingbeat and tail kinematics, as well as in body posture, to maintain flight control and stability. In heavy rain, hummingbirds increase their flapping frequency but reduce stroke amplitude, while simultaneously maintaining a horizontal body posture, and with lateral tail spreading exhibited by some individuals. Here, hummingbirds transiting a waterfall reduced stroke amplitude, but oriented the body and their tail more vertically, thereby reducing impact exposure to the falling water. Waterfall transit is, therefore, probably more dynamically challenging for hummingbirds relative to the perturbation of flight in heavy rain.

Although the flow rate (approx. 400 mm h$^{-1}$) used here was orders of magnitude lower than that of most natural waterfalls, it was greater than precipitation rates during extreme rainfall (greater than 50 mm h$^{-1}$; see [5]). Also, the water curtain used here was necessarily accelerating downward, at a Reynolds number based on the sheet thickness (approx. 3 mm) and speed (approx. 1.5 m s$^{-1}$) of about 4600, a value consistent with the turbulent and thus unsteady flow. Because actual waterfalls are much larger and more turbulent than the flow used here, we suggest that they represent scale-dependent barriers to small fliers; unsuccessful transit may also result in entrapment or injury at the base of the fluid structure. Various hypotheses have been proposed as to why some birds such as swifts often construct their nests behind waterfalls. The presence of waterfalls, with a high humidity and persistent entrained airflow, may result in a more constant microclimate which facilitates chick development and nest attachment on rock walls (see [4]). Alternatively, the largely inaccessible regions behind waterfalls may simply impede or preclude nest predation by volant and non-volant taxa alike. Large aerial nest predators might be able to transit a waterfall but would be challenged to manoeuvre and to land post-transit, we also suggest a third but related possibility: waterfalls may serve to exclude haematophagous fliers from nests. Nest parasites can not only elevate mortality rates of nestlings [19], but also may have long-term sublethal consequences for infested chicks. The selection of pristine nest sites can result in reduced parasite loads (see [20]). None of the aforementioned hypotheses are mutually exclusive, but relevant environmental and biological data are obviously difficult to obtain for this particular habitat. A study comparing ectoparasitic load between nestlings reared behind a waterfall and those reared in sites accessible to blood parasites is needed to evaluate this possibility. Nonetheless, flight through the water sheet used here clearly demonstrates scale-dependent success, which may indirectly underlie the tendency for some bird taxa to nest behind waterfalls.

Data accessibility. The datasets supporting this article have been uploaded as part of the electronic supplementary material.
Authors' contributions. V.M.O.-J. conceived of the study, designed the study, coordinated the study, carried out the laboratory work and statistical analyses, and drafted the manuscript. E.C.H. carried out the laboratory work, participated in the design of the study and critically revised the manuscript; M.S.L. carried out the laboratory work, participated in the design of the study and critically revised the manuscript. R.D. conceived of the study, designed the study, coordinated the study and drafted the manuscript. All authors gave final approval for publication and agree to be held accountable for the work performed therein.

Competing interests. We declare we have no competing interests

Funding. The author(s) received no specific funding for this work.

Acknowledgements. We thank members of the Berkeley Animal Flight Laboratory for various experimental insights, and Alejandro Rico-Guevara for specific recommendations on the text. This research was conducted in compliance with Animal Use Protocol 2016-02-8338 at the University of California, Berkeley CA, USA. Live bird trapping was carried out under permits from the United States Fish and Wildlife Service (MB054440-0) and the California Department of Fish and Wildlife (SC-6627).

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
