## [Reviewer comments · Royal Society Open Science]

Review History

Decision letter (RSOS-201185.R0)

Dear Dr Ortega-Jiménez

On behalf of the Editors, I am pleased to inform you that your Manuscript RSOS-201185 entitled "Natural Barriers: Waterfall Transit by Small Flying Animals" has been accepted for publication in Royal Society Open Science subject to minor revision in accordance with the referee suggestions.

The handling editors have recommended publication, but also suggest some minor revisions to your manuscript. Therefore, I invite you to respond to the comments and revise your manuscript.

- Ethics statement

- Data accessibility

<http://datadryad.org/submit?journalID=RSOS&manu=RSOS-201185>

- Competing interests

- Authors' contributions

- Acknowledgements

- Funding statement

Because the schedule for publication is very tight, it is a condition of publication that you submit the revised version of your manuscript before 26-Jul-2020. Please note that the revision deadline will expire at 00.00am on this date. If you do not think you will be able to meet this date please let me know immediately.

To revise your manuscript, log into <https://mc.manuscriptcentral.com/rsos> and enter your Author Centre, where you will find your manuscript title listed under "Manuscripts with

Decisions". Under "Actions," click on "Create a Revision." You will be unable to make your revisions on the originally submitted version of the manuscript. Instead, revise your manuscript and upload a new version through your Author Centre.

If your manuscript is newly submitted and subsequently accepted for publication, you will be asked to pay the article processing charge, unless you request a waiver and this is approved by Royal Society Publishing. You can find out more about the charges at <https://royalsocietypublishing.org/rsos/charges>. Should you have any queries, please contact openscience@royalsociety.org.

on behalf of Dr Jake Socha (Associate Editor) and Pete Smith (Subject Editor)
openscience@royalsociety.org

Associate Editor Comments to Author (Dr Jake Socha):

This study addresses how a few species of birds and insects deal with crossing through a waterfall in flight. It is both novel and interesting, and will make a strong contribution to the literature. The authors have sufficiently addressed the reviewer's comments from the previous submission to JRSI, and no further review is needed. However, I have one additional comment. As far as I'm aware, there are few studies on the mechanics of falling water on flying animals, but one prominent one was not mentioned. It would seem that some comparison to results in the 2012 PNAS paper by Dickerson et al. on how mosquitoes deal with raindrops would be highly relevant in the Discussion, particularly in regard to the physics of dealing with drops vs sheets. Please address this comment prior to publication.

Author's Response to Decision Letter for (RSOS-201185.R0)

See Appendix A.

Decision letter (RSOS-201185.R1)

Dear Dr Ortega-Jiménez,

It is a pleasure to accept your manuscript entitled "Natural Barriers: Waterfall Transit by Small Flying Animals" in its current form for publication in Royal Society Open Science.

You can expect to receive a proof of your article in the near future. Please contact the editorial office (openscience_proofs@royalsociety.org) and the production office (openscience@royalsociety.org) to let us know if you are likely to be away from e-mail contact -- if

you are going to be away, please nominate a co-author (if available) to manage the proofing process, and ensure they are copied into your email to the journal.

on behalf of Dr Jake Socha (Associate Editor) and Pete Smith (Subject Editor)
openscience@royalsociety.org

Appendix A

Dear Dr. Sanders,

I wish to resubmit an updated version of our accepted manuscript entitled "Natural Barriers: Waterfall Transit by Small Flying Animals". Please see below our response to the editor's comment. If you have any questions, please do not hesitate to contact me.

Sincerely,

Victor M. Ortega-Jimenez
Postdoctoral Researcher
Department of Ecology, Evolution, and Organismal Biology
Kennesaw State University

Comments to Author:

Editor:

This study addresses how a few species of birds and insects deal with crossing through a waterfall in flight. It is both novel and interesting, and will make a strong contribution to the literature. The authors have sufficiently addressed the reviewer's comments from the previous submission to JRSI, and no further review is needed. However, I have one additional comment. As far as I'm aware, there are few studies on the mechanics of falling water on flying animals, but one prominent one was not mentioned. It would seem that some comparison to results in the 2012 PNAS paper by Dickerson et al. on how mosquitoes deal with raindrops would be highly relevant in the Discussion, particularly in regard to the physics of dealing with drops vs sheets. Please address this comment prior to publication.

R. In a previous MS version we already included Dickerson et al (2012)'s paper in the discussion as follows. "The latter outcome agrees with experiments performed on mosquitoes exposed to falling droplets; impacts on their wings and legs can cause marked changes in body orientation, with a recovery time of ~100 ms [6]." However the citation was wrongly refereed as Andrew KD. et al. (2012). Thus, in the updated MS version that reference was corrected using Dickerson AK.